# Evolution of masting in plants is linked to investment in low tissue mortality

Valentin Journé [1] ✉, Andrew Hacket-Pain [2] & Michał Bogdziewicz [1] ✉

Masting, a variable and synchronized variation in reproductive effort is a prevalent strategy among perennial plants, but the factors leading to inter-specific differences in masting remain unclear. Here, we investigate inter-annual patterns of reproductive investment in 517 species of terrestrial perennial plants, including herbs, graminoids, shrubs, and trees. We place these patterns in the context of the plants' phylogeny, habitat, form and function. Our findings reveal that masting is widespread across the plant phylogeny. Nonetheless, reversion from masting to regular seed production is also common. While interannual variation in seed production is highest in temperate and boreal zones, our analysis controlling for environment and phylogeny indicates that masting is more frequent in species that invest in tissue longevity. Our modeling exposes masting-trait relationships that would otherwise remain hidden and provides large-scale evidence that the costs of delayed reproduction play a significant role in the evolution of variable reproduction in plants.

In perennial plants, reproduction can occur through spatially synchronized seed production, which varies substantially over time. In some years, investment in seed production is much higher than average, while in other years plants allocate few or no resources to reproduction, resulting in what is known as masting[1,2]. The concentration of reproduction in intermittent years appears heritable[3], and helps alleviate pollen limitation and reduce seed predation but comes at the cost of skipped reproductive opportunities[4–7]. The varying balance of masting costs and benefits is likely responsible for the rich diversity of reproductive behaviors observed in perennials, ranging from relatively regular fruiting to rare reproduction happening at long lags[1,8–11]. Large-scale variation in masting benefits is better explored compared to costs[1,9,11,12]. For example, interannual variation in seed production is high in the temperate zone, where the benefits of starving and satiating specialist seed predators are the greatest[1,13]. In contrast, the costs of missed reproductive opportunities have long been only theorized to be higher in species with high population growth rates and low adult survivorship[5,14], but this has remained challenging to test. Here, using trait-based approaches, we provide support for this central tenet of masting theory, showing

that masting predominately occurs in species with conservative plant tissues.

Accessible trait-based approaches can serve as indicators of life history strategies, aiding in the identification of functional constraints and trade-offs[15–18], and providing an avenue to investigate how varying costs of reproduction (skipped reproduction) shapes the evolution of masting. High stem tissue density (i.e. wood density) provides mechanical strength and reduces mortality, but limits growth rates, which distinguishes strategies reliant on stress persistence from rapid utilization of ephemeral opportunities[17]. We can thus expect stronger masting in species with high stem tissue density, as lower mortality rates due to stronger stress resistance should buffer against missed reproductive opportunities[14,19,20]. Similarly, productive but short-lived leaves with high nitrogen content and low leaf mass per area (LMA) are characteristic of cheap, acquisitive leaves that are efficient in resource-rich environments and associated with high population growth rates[20]. Such leaves should be thus associated with low interannual variation in reproduction[1,21]. In addition, high interannual variation should be also associated with large seeds if expensive reproduction strongly depletes resources after reproductive events[5,22]. Although these links

[1]Forest Biology Center, Institute of Environmental Biology, Faculty of Biology, Adam Mickiewicz University, Uniwersytetu Poznańskiego 6, 61-614 Poznan, Poland. [2]Department of Geography and Planning, School of Environmental Sciences, University of Liverpool, Liverpool, United Kingdom. ✉e-mail: journe.valentin@gmail.com; michalbogdziewicz@gmail.com

are theoretically established in the literature, supporting evidence is scarce, as data on seed production accumulate slowly and require significant investment[23,24].

The relationships between traits at large scales are complicated by their often-neglected direct (conditional) and indirect (marginal) relationships[25,26], through the intricate connection of climate, geography, or phylogeny. In the case of masting, stem tissue density tends to be high in the tropics where interannual variation in seed production is low[9,17]. Therefore, a negative correlation between interannual variation in seed production and stem tissue density could be an indirect relationship resulting from latitudinal covariance in these traits. Alternatively, the relationship could be direct if the low interannual variation in seed production requires species to produce conservative stems. Indirect relationships may also arise from phylogenetic conservatism. Certain taxa may exhibit large interannual variations in seed production and high stem tissue density even if environmental conditions that select one or both traits change. Traditional summaries such as principal component analysis (PCA) summarize correlations that include all the indirect ways traits could be associated[26,27]. To address this issue, novel methods such as joint attribute modeling enable the decomposition of relationships into direct and indirect, driven by either climate or phylogeny[26,28]. These statistical tools synergize with the recent advancement of global coordination in monitoring and seed production data synthesis, allowing tests of decades-old assumptions of the field while accounting for longstanding issues with covariance between variables.

In this study, we explore the relationship between masting, phylogeny, climate, and functional diversity across 517 species of vascular plants, including herbs, graminoids, shrubs, and trees from various biomes (Fig. 1). We use MASTREE+, a database that provides information on annual variations in plant reproductive effort[24]. We characterize the variability of seed production in each species using two commonly used masting metrics, the coefficient of variation (CV), and the lag-1 temporal autocorrelation (AR1), which describes the tendency of high seed production years to be followed by low seed production[1,29]. Using joint attribute modeling, we extract conditional relationships driven by climate and phylogeny and associate large interannual variation in seed production with a need for conservative

tissues. This provides large-scale evidence that the costs of delayed reproduction play a significant role in the evolution of variable reproduction.

## Results

### Masting on the spectrum of plant form

We start with results derived from the traditional principal component analysis (PCA) approach to illustrate the challenges associated with mixing conditional and marginal relationships. Principal component analysis of functional traits and masting metrics indicates that masting is largely independent of functional traits. The PCA of six functional traits and masting metrics indicated that the 517 species examined here had two primary sources of variation: an axis of leaf economics (Axis 1: leaf mass per area, leaf nitrogen, leaf area) and plant size (Axis 2: seed mass, plant height, and stem tissue density), with no contributions from masting metrics (i.e. coefficient of variation, CV, and the lag-1 of temporal auto-correlation, AR1 of seed production). Instead, masting generated a distinct axis of variation (Axis 3), with species exhibiting high CV and negative temporal autocorrelation of seed production concentrated at one end of the axis (Fig. 2 & Fig. S1). However, the correlation summary mixed conditional and marginal relationships conferred by phylogeny and climate, which each had strong effects on masting, as explained below.

### Masting on the Tree of Life of plants

The coefficient of variation (CV) and the lag-1 temporal auto-correlation (AR1) exhibited phylogenetic coherence, with CV coherence being about twice as strong (CV: $\lambda = 0.48$, $p < 0.0001$; AR1: $\lambda = 0.27$, $p < 0.0001$, as shown in Fig. 3 and Fig. S3). Several groups were found to have a high concentration of species with a very high coefficient of variation in seed production (Fig. 3). These groups included Poales' *Chionochloa* and *Miscanthus*. The Pinales order also included high-CV genera such as *Abies*, *Juniperus*, and *Picea*, as well as mixed ones such as *Pinus*. Fagales were also mixed, including high-variability genera such as *Betulaceae* and mixed ones such as *Fagaceae*, which had high-CV *Fagus* and diverse *Quercus*. Low CV was common in Magnoliales, Gentianales, and some genera of Cornales and Malvales, such as *Cistaceae* and *Cornaceae*. Highly negative temporal autocorrelation of

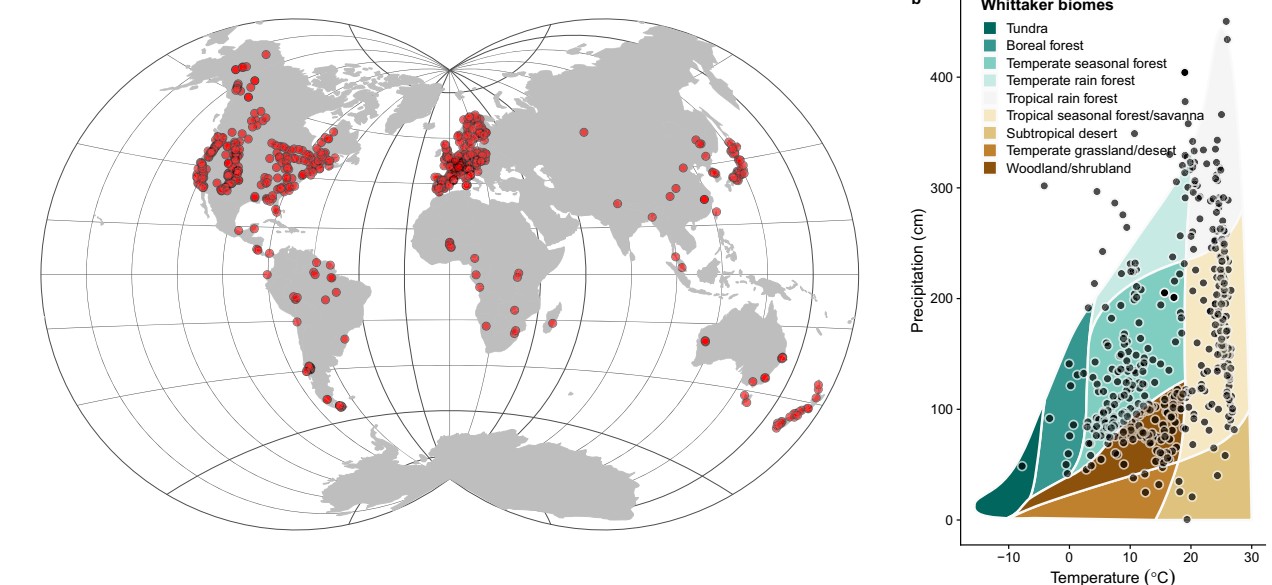

**Fig. 1 | MASTREE+ sites used in the analysis, and climatic space for the species analyzed. a** Location of MASTREE+ sites (red dots) included in this study (data displayed in Van der Grinten IV projection). **b** Climatic distribution of our sites. Each dot represents average climatic conditions (mean annual temperature, MAT, and mean annual precipitation, MAP) at the species distribution level (n = 517 species). Data on species distribution was largely derived from the Global Biodiversity Information Facility (GBIF, www.gbif.org) (see Methods). The Whittaker biome plot is included in the background for context.

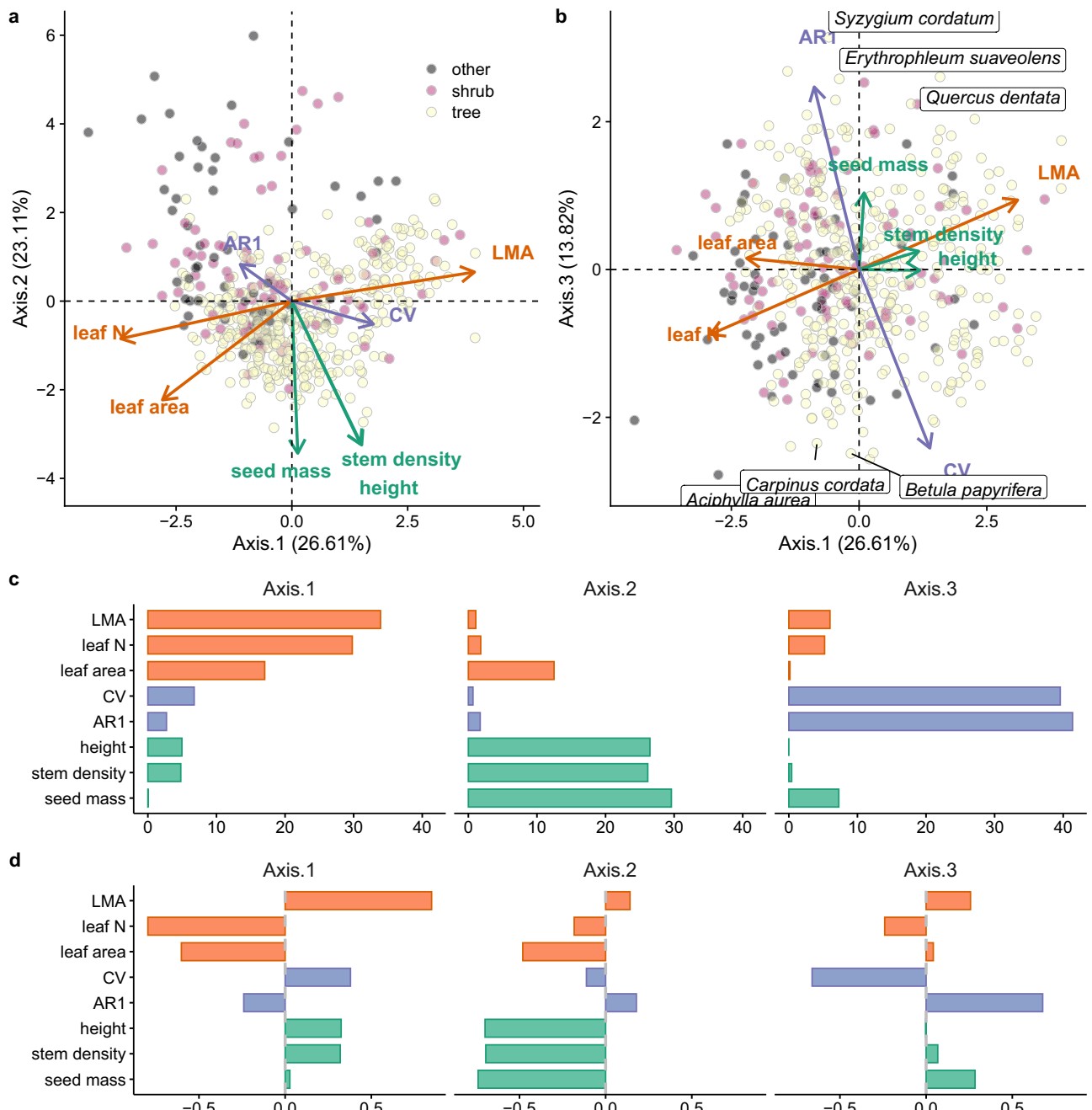

**Fig. 2 | Masting metrics (coefficient of variation, CV, and lag-1 temporal auto-correlation of seed production, AR1) on the spectrum of plant functional traits. a** Biplot of principal components that summarized axes 1 and 2, and (**b**) axes 1 and 3. The PCA included plant functional traits (stem tissue density, leaf area, leaf nitrogen, leaf mass per area LMA, plant height, and seed mass) and masting metrics (CV and AR1). Arrow length indicates the loading of each considered trait onto the axes. Points represent the position of species color-coded according to their growth form (yellow for trees, purple for shrubs, and gray for others that included graminoid and non-graminoid herbaceous and climbers). **c** Summary of PCA loadings, and (**d**) contributions to the three axes of variation. The bars at (**c**) and (**d**) are color-coded to match the colors of axes (at **a**, **b**) to which the traits loaded the most. The trait probability density function is given in Fig. S1, and CV/AR1 by growth form with PCA in Fig. S2.

seed production was a characteristic trait of Fagales (Fig. S3). Other groups, such as Rosales or Pinales, were mixed, while Malpighiales, Gentianales, and Magnoliales were dominated by positive autocorrelation.

### Masting across climates
Although interannual variation (CV) and lag-1 temporal auto-correlation (AR1) of seed production were not correlated (Fig. S5), they responded to the climate in opposite ways that resulted in a convergence of high CV and negative AR1 in the same climates (Fig. 4). Positive temporal autocorrelation was observed in species that grow in hot and dry environments, such as subtropical deserts or tropical seasonal forests (Fig. S6), where low CV was also common (Fig. S6). Conversely, negative AR1 and high CV were predicted in temperate and boreal forests, which are characterized by intermediate annual temperatures and precipitation (Fig. 4). We also explored models that were supplemented with climate variability (standard deviation of the monthly mean temperatures and coefficient of variation of the

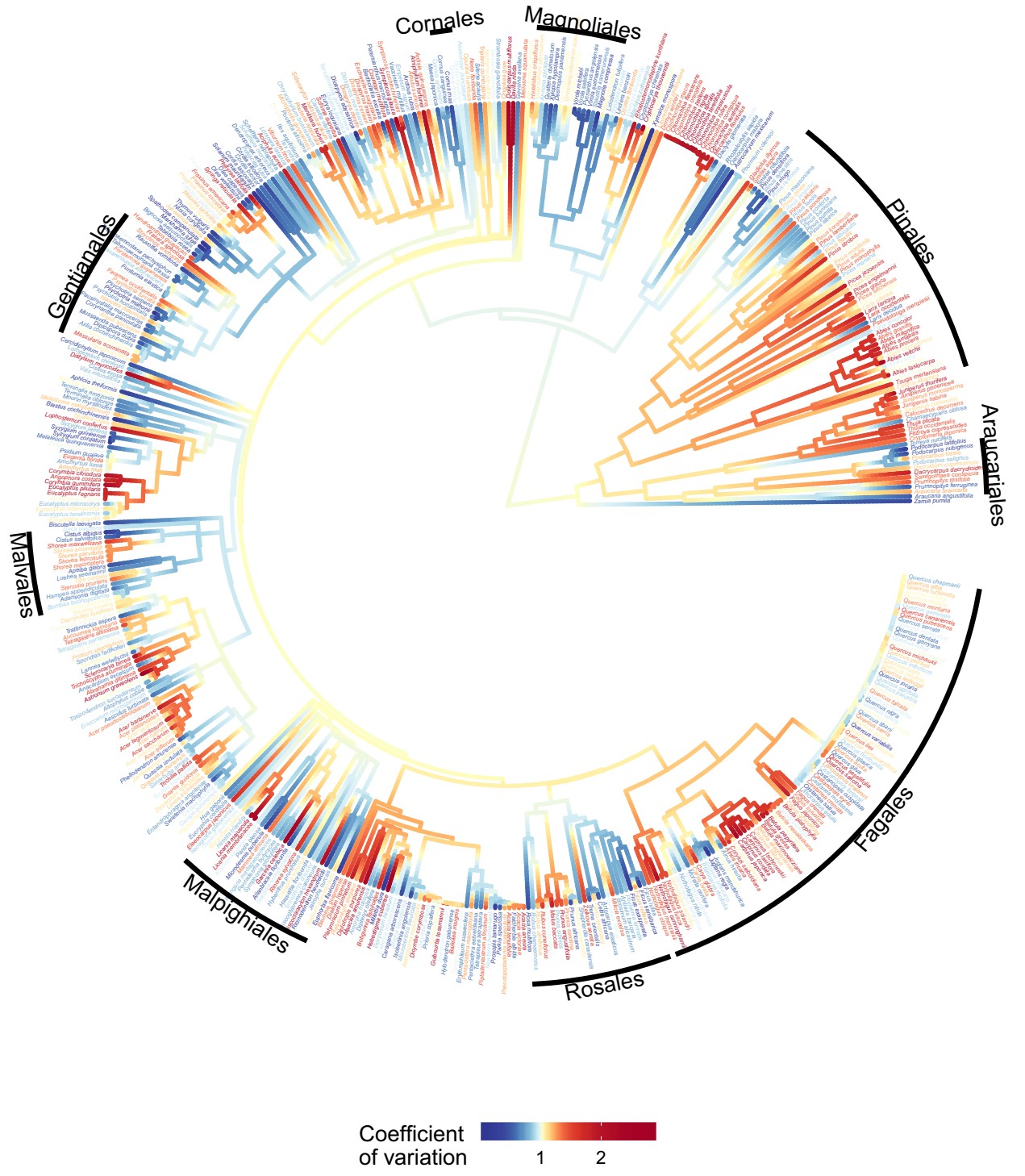

**Fig. 3 | Coefficient of variation of seed production mapped onto a plant phylogeny.** Warmer colors (reds) indicate higher, while blue lower CV (the phylogenetic signal is calculated using Pagel's $\lambda = 0.48$, $p < 0.0001$, $n = 518$ species). Distributions of the masting metrics are in Fig. S4. Orders of plants are provided at the periphery of the phylogenetic tree.

monthly precipitation), but the inclusion of climate variability has not improved our model's fit (Table S2).

**Masting and traits, accounted for climate and phylogeny**
The conditional prediction from generalized joint attribute modeling (GJAM), which accounted for the effects of phylogeny and species climatic niche on masting, revealed that species with dense tissue stems and conservative leaves characterized by high mass per

area tend to have higher coefficients of variation in seed production (Fig. 5). There was also a weak (non-significant in the full model) association between high CV and small seeds (Fig. 5, Table S1). These effects suggest that correlations (or the lack thereof) observed by PCA between traits and masting metrics were mainly driven by climate or shared ancestry. For instance, stem tissue density is highest in climates where masting is lowest (Fig. S7), but this negative covariance changes sign once the climate is taken into account.

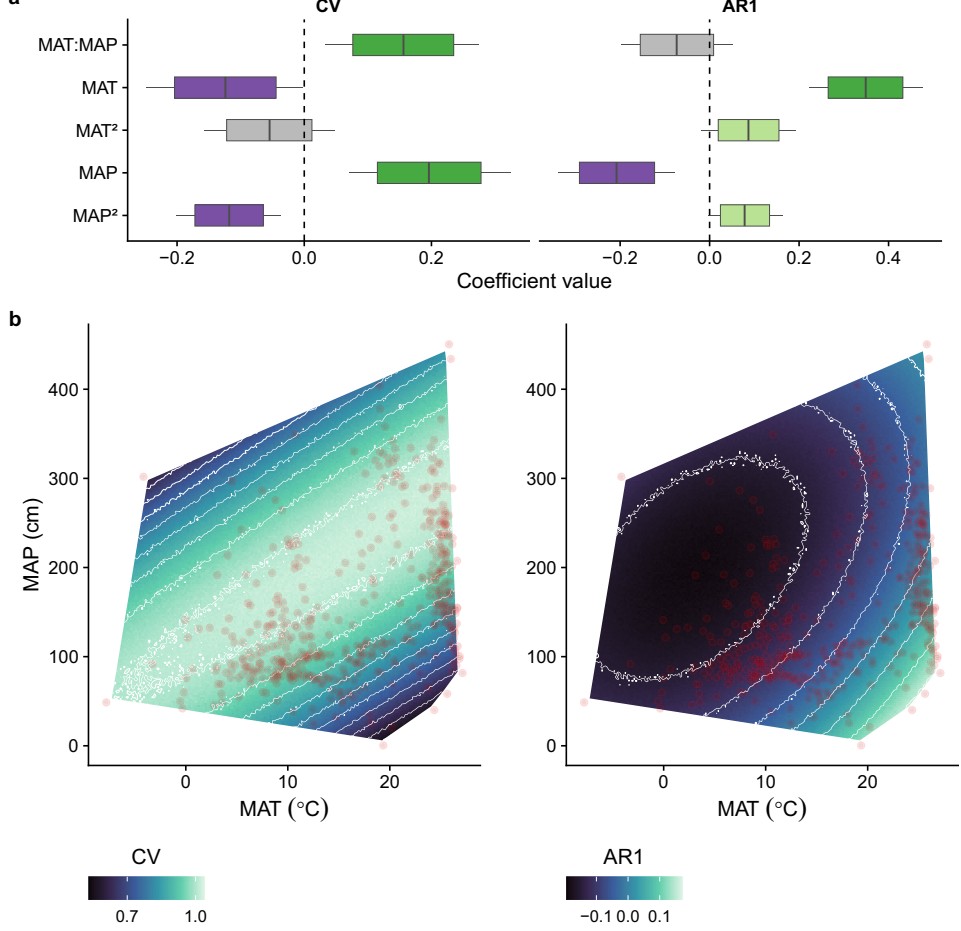

**Fig. 4 | Summary of climate effects on masting, derived from the GJAM model that included coefficient of variation (CV) and temporal autocorrelation (AR1) as responses (*n* = 517 species). a** Boxplots show the marginal posterior distributions of the GJAM-derived coefficients. Specifically, boxes show mean effect size as vertical lines and are bounded by 80% credible intervals (CI), with 95% CI as whiskers. Colors highlight signs of the correlation (green for positive and purple for negative), with opacity increasing from 80% to 95% of the distribution outside of zero. Gray indicates coefficients that overlap zero. **b** Effects of mean annual temperature (MAP, in *°C*) and mean annual precipitation (MAT, in cm) on CV and AR1. The surface shows the conditional relationship between CV/AR1 and MAT across levels of MAP. Convex hull is defined by species observations (red dots). MAT and MAP are defined for each species' distribution derived from the Global Biodiversity Information Facility (GBIF, www.gbif.org). Biplots of relationships between CV/AR1 and MAT and MAP are in Fig. S6. Climate effects on functional traits are in Fig. S7.

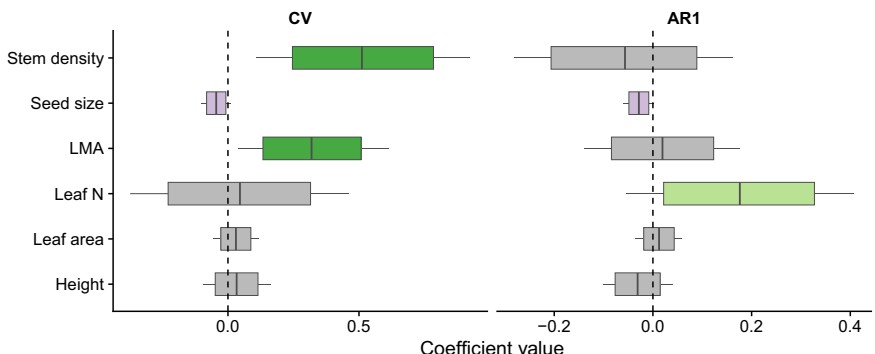

**Fig. 5 | GJAM-derived, conditional relationship between masting metrics (CV and AR1) and functional traits (stem tissue density, seed mass, LMA, leaf N, leaf area, and plant height) after accounting for effects of climate and phylogeny (*n* = 517 species).** Boxplots show the marginal posterior distributions of the GJAM-derived coefficients. Specifically, boxes show mean effect size as vertical lines and are bounded by 80% credible intervals (CI), with 95% CI as whiskers. Colors highlight signs of the correlation (green for positive and purple for negative), with opacity increasing from 80% to 95% of the distribution outside of zero. Gray indicates coefficients that overlap zero.

## Discussion

Interannual variation in seed production across 517 species is associated with restricted climatic and phylogenetic space and conservative tissues that include higher stem tissue (wood) density and higher LMA. First, the coefficient of variation of seed production was highest in temperate and boreal climates, which supports previous studies that have shown the CV to be highest at mid-latitudes[1,9]. Second, masting has evolved multiple times across the Tree of Life in plants, in growth forms ranging from grasses to trees. Nonetheless, numerous branches have split into high and low-variability groups, perhaps because species quickly lose their inherited seed production variability once there is no ongoing selection for it (e.g., low seed predation or high pollination can be achieved via other routes). Third, high interannual variation in seed production is concentrated in life history strategies that invest in low mortality. High survival rates decrease the costs of missed reproductive opportunities, which is a major masting cost that can prohibit masting evolution even when there is a strong selection for it[7,19]. Thus, costs of delayed reproduction appear a major factor driving the evolution of masting across species.

Masting is a widespread phenomenon in the Tree of Life of plants. Although the coefficient of variation (CV) of seed production exhibited relatively strong phylogenetic coherence, branches lacking closely related species that have reverted from masting to regular seed production were rare. For instance, the *Betulaceae* family, comprising *Betula*, *Carpinus*, and *Alnus*, displayed generally high variability, with exceptions including *Alnus hirsuta* and *Betula pendula*. The closely related *Chionochloa* species all showed highly interannually variable seeding patterns, with related *Dactylis glomerata* being a low-variability exception. Perhaps the high costs of high seed production variability mean that if the need for masting (e.g., high seed predation rates or low pollination efficiency) can be circumvented through less costly alternatives, regular seeding re-evolves. In this context, oaks represent a notable example of diversity and rapid transitions between low and highly variable strategies, contrasting with Pinales, where masting was lost less frequently. A comparison of these two groups to understand why masting is almost always beneficial in Pinales, such as *Picea* or *Abies*, but can quickly cease to be so in *Quercus*, is a promising area for future research. Are the costs of masting systematically smaller in Pinales, or is the need for masting (e.g., low pollination efficiency) systematically greater? One interesting way forward is to examine this question in light of the high resprouting abilities of oaks but not pines[30].

A high coefficient of variation in seed production does not necessarily imply a need for negative lag-1 temporal autocorrelation, indicating that the two can evolve independently[9,10]. High CV values without strongly negative AR1 may happen if mast years are not followed by complete failure years[9,10]. However, climate effects on these metrics lead to the convergence of high CV and highly negative AR1 in the same boreal and temperate habitats. Predator satiation is most effective at mid-latitudes[13], which is often explained by a lower diversity of alternate food resources for seed consumers that helps control their populations[1,9]. Thus, the high potential effectiveness of predator satiation may lead to stronger selection for both high CV and negative AR1 in such biomes. Alternatively, species in the boreal and temperate zones may rely less on mutualistic interactions[31], which tend to select against masting[1,11,32]. For example, wind pollination is less frequent at low latitudes[33], and the absence of negative AR1 may help avoid the starvation of animal pollinators in these systems. Finally, to the extent that negative AR1 reflects resource depletion following high-seeding years[34,35], convergence between high CV and negative AR1 could be driven by stronger resource constraints in certain climates[21]. Irrespective of the reason, in climates where high CV and negative AR1 co-occur, masting-driven pulsed resources would be expected to involve frequent famines[36,37], creating an especially unstable base of food webs in these biomes.

Masting is associated with a restricted functional trait space. High interannual variation in seed production is common in species with high stem tissue density and, to a lesser extent, in species with high leaf mass per area (LMA). These species invest heavily in constructing tissues, resulting in slower returns on nutrient investment but higher survival through higher defenses against physical damage and herbivores[17,38,39]. Theoretical models suggest that the significant costs of missed reproductive opportunities can prevent the evolution of masting, even in the presence of significant benefits such as improved pollination and reduced seed predation[7,14,19]. In this context, our results support this long-standing theory, testing of which has previously been frustrated by lack of data. What is more, recent studies suggested that the other theoretical masting cost, negative density-dependent seedling survival[5,40], may be lower than expected. Theory predicts that negative density-dependent seedling survival can prohibit the evolution of masting in plants that have high adult survival[40]. However, recent evidence implies that masting does not result in lowered seedling survival in *Sorbus aucuparia*[12], and may even increase seeding survival in tropical communities[41]. Generally, negative density dependence appears fairly weak on average and highly variable among species, suggesting that its generality may be overstated[42]. Together with our results, these suggest that the costs of delayed reproduction may be a major mechanism driving the evolution of masting across plant life history strategies.

We also found no support for theories linking high CV with large seed[5,22]. We speculate that the tendency for high CV in small-seeded species, in contrast to theoretical predictions, may result from contrasting selection pressures. For example, small seeds are correlated with seed bank persistence in the soil[43], which is another way to circumvent the costs of missed reproductive opportunities[19]. Consequently, if there are ways that small-seeded species can reduce the costs of missed reproduction, masting might evolve more readily, offsetting the expected direct effect of large seeds on masting.

In summary, our analysis supports the idea that the extent of year-to-year variation in masting is regulated by a species' phylogeny, location (climate), and life history (plant form). The effects of climate and phylogeny on mast seeding and functional traits necessitated conditional predictions that extracted direct associations[27,28]. A PCA analysis that combined all the ways in which variables can be linked suggested that masting created a third, mostly independent dimension of variation in plant traits. This outcome would support a twin-filter model, according to which primary strategies, such as the fast-slow leaf economics spectrum[44], determine plant persistence for climate and habitat norms, whereas traits involved in episodic events, including reproduction, affect fitness regardless of other traits[45]. In other words, masting would evolve whenever there is a need for it, regardless of the plant form. However, by extracting direct effects, we showed that links among traits and variation in seed production were concealed by their covariance with climate and phylogeny. That modeling reversed the analytical outcomes, showing that the costs of delayed reproduction may prevent masting in fast-growing, low-survival plant forms. The required next step is to directly link masting with life history traits (population growth rate, size at sexual maturity, mortality rates) which, with growing data availability[46], may soon become feasible.

## Methods

### Data description

Our analysis is based on MASTREE+, a database of annual records of population-level reproductive effort of 974 from all vegetated continents[24]. For our analysis, we excluded time series that were on an ordinal scale and those based on pollen measurements. We analyzed two subsets of the data. One, broader, was limited to time series with at least 5 years of observations. That analysis is reported in the main text. Second, a more restrictive analysis included time series with at least

10 years of records. Results of that analysis are reported in the Supplementary Section and provide quantitatively the same outcomes.

**Masting metrics.** We computed the coefficient of variation (CV, standard deviation divided by mean of seed production) for each site-species combination. The CV is commonly used in masting studies to describe inter-annual variations of seed production[1,29,47]. We also computed lag-1 temporal auto-correlation of seed production (AR1), which characterized the tendency of high-seeding years to be followed by low-seeding years. For each species, we computed the average CV and AR1. To compute auto-regressive correlation we used the `acf` function in R[48].

**Functional traits.** We extracted species-level functional traits from[49], which include Leaf Mass Area (LMA, in g.m$^{-2}$), stem tissue density (SSD, in mg.mm$^{-3}$), plant height (ph, in m), leaf nitrogen (ln, in mg.g$^{-1}$), seed size (sm, in mg), and leaf area (la, in mm$^{-2}$), and plant growth form (Fig. S8). We obtained plant growth form, which includes trees, shrubs, and other categories, with graminoid and non-graminoid herbaceous, and climbers (see distribution in Fig. S2).

Full trait information obtained from[49] was available for 210 species from MASTREE+ database. To increase species coverage, we performed a trait-imputation procedure. We used machine learning that accounted for species phylogeny[50,51]. We filled only species that had information for at least three functional traits (out of six used in the analysis)[18,50]. First, we log10 transformed known functional traits and incorporated phylogenetic information for each species[52]. The phylogenetic information was summarized by eigenvectors extracted from a principal coordinate analysis (PCoA), which represented the variation in the phylogenetic distances among species. We used the first ten axes of PCoA for the imputation process[50,52]. The phylogeny was obtained using the R package `V.Phylomaker2`[53,54], with the `GBOTB.extented.TPL` tree as a backbone[55,56], and scenario S3 to generate the phylogeny[54,57]. Imputation of missing trait information with machine learning has been done through the R package `missForest`[58]. That imputation allowed us to increase the sample size (i.e. species for which we had full traits and seed production data) to 517 species. The GJAM model without trait data imputation generated qualitatively similar results for CV (Table S1). In the case of AR1, lack of trait imputation resulted in a positive association between leaf N and AR1, and a negative between height and AR1 being significant. That hints that acquisitive leaves may buffer against strong post-mast seeding failure[21], although it is unclear why smaller plants have more negative AR1. For consistency, we discuss only the results with the data imputation in the main text.

**Abiotic variables.** We determined the species' climatic niche by using species occurrences extracted from Global Biodiversity Information Facility (GBIF, www.gbif.org) through the `rgbif` package[59] (data request: https://doi.org/10.15468/dl.jxyrhk)[60]. We removed species occurrences from GBIF that are incorrectly or vaguely reported and outliers by using the R package `CoordinateCleaner`[61] to keep precise species locations (mean number of occurrences for our species = 7609, CI975 = [1; 105,093]). Next, for each occurrence, we extracted a mean annual temperature (MAT, in °C) and mean annual cumulative precipitation (MAP, in cm) by using CHELSA data[62], and averaged those values from all occurrences per species to one value per species range (MAT and MAP). For each species, we used average species climatic conditions from MASTREE+ if the number of sample sites from MASTREE+ was higher than the number of species occurrences from GBIF ($n$ = 55 species). We used GBIF-based climate to accommodate functional traits and masting metrics at species-wide averages. Nonetheless, MAT and MAP obtained through MASTREE+ sites and GBIF present strong correlations (Fig. S9), and using both provides qualitatively the same results.

## Analysis

**Phylogenetic analysis.** We estimated the phylogenetic signal of the coefficient of variation (CV) and temporal auto-correlation (AR1) of seed production with Pagel's $\lambda$[63]. Pagel's $\lambda$ is based on the Brownian Motion evolutionary model and ranges from 0, when there is no phylogenetic signal, to 1 where the phylogenetic signal is estimated to be very strong. The Pagel's $\lambda$ was estimated by using the `phyolosig` function from `phytools` R package[64] and visualized with `ggtree`[65]. We used a plant phylogenetic tree provided by[55].

**Multivariate analysis.** We used the principal component analysis (PCA) to describe the multivariate trait spectrum, which included the six functional traits and two masting metrics (CV and AR1). We kept functional traits log10 transformed. We standardized and centered variables. We used `ade4`[66] R package. Moreover, we estimated the occurrence probability of trait combination in two-dimensional space (determined by the PCA axis 1 and 2, or by axis 2 and 3) with their bivariate trait combinations. We used the two-dimensional kernel density estimation and determined the highest probability trait occurrence[18,51].

**Joint model analysis.** We jointly modeled functional traits and masting metrics using the generalized joint attribute modeling (GJAM,[28]). Average climatic conditions per species range (occurrences obtained via GBIF, see above) were included as predictors, i.e. mean annual temperature (MAT) and mean annual precipitation (MAP). We tested a set of models with different combinations of the interaction between MAP and MAT, and their quadratic terms. Model selection was based on the Deviance information criterion (DIC). The GJAM allowed us to accommodate the dependence between traits and phylogeny as random groups. To this end, we followed past studies that used a similar approach[27,67], and grouped species according to genus or family (when the genus had <10 species). We used the 'multiple' category for families with <5 species.

We accommodated the mutual dependence structure of traits and isolated their effect on masting metrics through conditional prediction[27,68]. Conditional prediction offers an estimation of the relationships between traits and masting metrics while accounting for the effects that come through climate and phylogeny. These conditional parameters are obtained via `gjam` R package[28], by specifying traits being conditioned (here, functional traits) on the variable of interest (here, CV and AR1 of seed production). In doing this, we first estimate how responses (functional traits and masting metrics) correlate with climate. Next, the relationships among responses are estimated, after accounting for the predictors (climate and phylogeny). The `gjam` is an open-access R package gjam available on CRAN.

### Reporting summary
Further information on research design is available in the Nature Portfolio Reporting Summary linked to this article.

## Data availability
The data used in this study have been deposited in the Open Science Framework (OSF) (https://osf.io/57w2q/). The full MASTREE+ dataset is available in[24]. Traits have been downloaded from[49]. Climate data have been extracted from CHELSA at https://chelsa-climate.org/.

## Code availability
R statistical software v4.3.0 was used in this work[48]. All analyses used published R packages.

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

## Acknowledgements

We thank Jessie Foest for her help with extracting data from the MAS-TREE+ database and Kevin Sartori for early discussions and suggestions. This study was conceived during a workshop funded by the UK Natural Environment Research Council grant no. NE/S007857/1, and uses a dataset created as part of that project. VJ was supported by project No. 2021/43/P/NZ8/01209 co-funded by the Polish National Science Centre and the EU H2020 research and innovation program under the MSCA GA No. 945339. MB was supported by the European Union (ERC, Forest-Future, 101039066). Views and opinions expressed are however those of the authors only and do not necessarily reflect those of the European Union or the European Research Council. Neither the European Union nor the granting authority can be held responsible for them.

## Author contributions

V.J. and M.B designed the study. V.J. led the analysis with inputs from M.B. and A. H-P. M.B. and V.J. co-wrote the first draft of the paper. Revisions were done by all Authors.

## Competing interests

The authors declare no competing interests
