## [Peer Review File · Nature Communications]

Evolution of masting in plants is linked to investment in low tissue mortalityReviewers' Comments:

Reviewer #1:

General

In this study the authors compile a large dataset of plant functional traits, including masting characteristics, and climate to investigate patterns of masting across vascular plants. The dataset is interesting and understanding why some species exhibit masting and not others is an intriguing and on-going research question. However, it wasn't clear from the set up of this paper what was novel about their study. It wasn't clear why the authors were doing what they were doing. It read more like a data exploration than a novel study (the authors followed the exact methods they previously published with a new dataset, which is not bad but I wish the study was presented to us as a novel exciting study addressing new questions).

Besides the way the study is presented, I do have some questions/issues with the method they follow. The first big concern I have is the fact that they only have functional traits for 210 species and had to use machine learning and phylogeny to infer missing data for 307 species. This is problematic to me for a few reasons. 1. Not all traits are phylogenetically conserved... 2. They have more missing data than they have data.. 3. It seems that at the end the study ends up being circular.

The second question I have about the method is in the choice of their climatic variables. Using MAT and MAP is easy but is it sufficient? When studying a temporal phenomenon like masting, why aren't any climate variability measures included in their models? Obviously the costs and benefits of masting will greatly differ in a constant environment vs. a highly variable one.

Finally, I made several comments throughout the figures which could be greatly improved. I am also confused about the caption and what the figures are showing (for example in Figure 5. caption explains the whiskers are the 95% CI but then some posterior distributions that clearly overlap with 0 are dimmed significant...). This is pretty confusing and makes me doubt the results overall.

In conclusion, I think there is some potential with this study but the authors need to present their work and their results in a more convincing ways and rethink (or justify better) some of their analyses/choices.

Introduction

Overall, the introduction doesn't do a very good job setting up the novelty of this and why it is an important study even.

Results

Since 517 species were analyzed, it would have been interesting to know about variation within the broad life form. For example, comparing results among trees, shrubs, grasses as presented the results are a little too superficial and I would have liked a deeper analyses of the results. Line 99. Need to specify what this principal component analyses was (PCA of X functional traits and Y masting metrics).

Lines 99-114. I am not sure why the authors bother with a simple PCA rather than focusing on the results from GJAM. Need to justify the PCA approach. Also, the authors should spend more time exploring results with GJAM (See my first comment in the results).

Figure 1. The Whittaker diagram is a little strange with many of your points falling outside of the biomes. Use a different diagram? Double check the T and P values of some of these locations? Color code the map with biomes instead in which case you don't need b at all.

Figure 2. Add a clip-art of a species representative of the two extreme masting strategies. Change the point colors (it is too similar to the color picked for the three axes)

Instead of plotting Axis.3. vs Axis.2, Plot Axis. 3 vs. Axis.1 so a and b and easily be interpreted together.

Figure 4. Now sure why here you are showing 97.5 CI but then in Figure 5 you can showing 95% CI. I'd be better to remain consistent across results.

Figure 5. Besides a different CI than in Figure 4, I am confused about some of the results. If the whiskers extend to 95%CI, then seed size is not significantly associated with CV or AR1, and leaf N is not significantly associated with AR1.

Methods

Lines 254-269. You obtained data for 210 species, and used a machine learning algorithm to get data for 517 additional species using phylogeny. This is worrisome to me for several reasons: 1. Many functional traits are not phylogenetically conserved, 2. You end up creating data from more species than you have data for, 3. Then your analyses become circular (phylogeny use to predict traits, then traits and phylogeny use to predict masting after removing climate...). This issue needs to be addressed.

Lines 271. Is it unclear why the authors decided to focus on mean annual T and P only. What about variability? Masting is temporal phenomenon and variability in the climate should be considered.

Discussion

Line 213. Again you didn't really tackle the idea of growth form. You have the data but didn't explore it, you stopped at functional traits.

Lines 217-222. How complicated would it be to test this evolutionary model? Could you add it to this paper to make this study stronger?

Reviewer #2:

Remarks to the Author:

The manuscript tests important hypotheses on the evolution of masting in perennial plants. The data and analyses reported are robust, and their results present a mix of confirmations and novel evidence on traditional and so far untested theories. The paper therefore makes an important, although preliminary contribution towards the understanding of the evolution of masting.

My only main concern stems from one of the predictors considered, i.e., climate. The question is simple: how can we be sure that current climatologies (extracted by CHELSA) are the same one that acted upon evolution? Also given that climate stability has been higher or lower in different parts of the world (as influenced by eg glaciation, volcanism, plate tectonics...). While the temporal scale for other predictors (phylogeny, life traits) is consistent with evolutionary dynamics, I am afraid of a potential mismatch between these and the climate information used in the paper. I wonder whether using some palaeoclimate data (eg from the PAGES project, or similar) might be useful to answer this question.

Minor comments follow:

1 Conservative tissue construction: not entirely clear from the title what this is about - how about "the need for conservative tissue"?

15 extreme: not sure this is a proper descriptor for masting... I suggest to stick with "variable"

16 perennials -> perennial plants

19 ancestry -> phylogeny? evolutionary history? (also at L213)

20 was rare -> is rare

20-21 it is rare... to lack: replace two negatives with one assertive sentence, e.g., it is common... to include

23: was -> is

23: invest in low tissue mortality: not entirely clear... do you refer to species whose tissues are predominantly conserved from year to year (hence the title)? I think this trait is not so familiar to most readers, so a sentence explaining why it was studied and why it is important might benefit clarity

43: remove first names in citations

50 and rapid utilization: maybe "from" rapid utilization?

51 such as -> measured by

63-64: not entirely clear why high stem tissue density would be related to lower mortality rates (eg add ..."due to stronger stress resistance")

69 and elsewhere: I prefer "stem tissue density" to just "stem density", which might be confused with the number of stems per unit area

73 "if high interannual variation in seed production requires species to produce conservative stems": but the evidence points to the opposite, ie. little masting in dense tropical woods

fig 1: why some dots are well out of existing Whittaker biome space?

101: seed mass, tree height and stem (tissue) density can also be expected to vary on separate axes, eg, tall, fast-growing and light-wooded, light-seeded trees. Was this observed on other PCs?

151 "once there is less ecological need for it": losing an evolutionary trait should imply something more than just "no need" for it (in which case it could persist by evolutionary inertia), i.e., it becomes too costly for the new conditions and/or discourages survival and effective reproduction

159-160: this paragraph takes for granted that masting, likely a multi-genetic and very complex trait, is strongly inherited. Authors should make this assumption explicit alongside with appropriate references to support it

170-2: oaks are sprouters while pines are not. Maybe there could be a role for disturbance resilience here? see <https://royalsocietypublishing.org/doi/full/10.1098/rstb.2020.0384>

173-74: that CV and AR1 are not necessarily linked is new to me. This is an interesting conclusion, maybe authors should elaborate a bit more how and why there can be a high CV without AR1 (no depletion?), or even weirder, a high AR1 with small CV (what would that mean from a mathematical point of view?)

207-211: also, the importance of post-germination survival could be invoked here (eg for species with large seeds that are subject to a large survival pressure in the seedling stage)

212-26: In this "summary", novel information and interesting lessons to be learned are presented, but the text is quite compact. I suggest expanding a bit in a full Conclusion paragraph, so as to have a bit more room to explain implications properly.

255+: I am not an expert in this technique. However, would carrying out the analysis on only the 210 species with complete traits have changed the main results?

277+: how about testing also extreme temperatures and precipitation rather than their averages, as an indicator of potential environmental stress or disturbances? (as per hypotheses by <https://royalsocietypublishing.org/doi/full/10.1098/rstb.2020.0384>)

278: what is the time window used by CHELSA climatologies? Surely this is not at all comparable to the kind of time span required for selective pressure? In other words, how can we be sure that these are the climates that acted upon evolution? Also given that climate stability has been higher or lower in different parts of the world (as influenced by eg glaciation, volcanism, plate tectonics...)

279 aggregated: you mean by averaging?

280-82: this procedure might be unnecessary, or even accurate, if the observations from mastree would cover a smaller territory than the observations from GBIF, although fewer

298+: PCA is best for linear relationships, and can be affected by data distortion (horseshoe effect). Have you thought of running a NMDS to account for nonlinear effects, and/or DCA to correct for the arch distortion?

REVIEWER COMMENTS TO AUTHOR

Referee: 1

General

In this study the authors compile a large dataset of plant functional traits, including masting characteristics, and climate to investigate patterns of masting across vascular plants. The dataset is interesting and understanding why some species exhibit masting and not others is an intriguing and on-going research question. However, it wasn't clear from the set up of this paper what was novel about their study. It wasn't clear why the authors were doing what they were doing. It read more like a data exploration than a novel study (the authors followed the exact methods they previously published with a new dataset, which is not bad but I wish the study was presented to us as a novel exciting study addressing new questions).

1.1. Thank you for that important comment. We believe that the key finding of our work is that variation of seed production correlates with functional traits associated with low mortality. The largest masting cost is skipped reproductive opportunities, at the cost of each missed opportunity should decline as the annual survival increases. That is a fundamental assumption of masting theory that has not been tested at scale. We filled that gap using novel methods that allowed separation of the effects of climate. We believe that the non-standard modeling is key as well, since traditional PCA run on such large scales is confounded by climate heterogeneity.

We have rewritten the Abstract, Introduction, and Discussion of the paper to more directly flesh-out the novel results the study brings. This include revisions of:

- Abstract; highlight of the novelty (masting costs, novel methods), L23-L28
- 1st paragraph of the Introduction, highlight the novelty (masting benefits better explored then costs, we fill that gap), L39 - L48
- 2nd paragraph on traits; we shortened it, focusing directly on the masting-trait relationships of high relevance in the current context
- 3rd paragraph on conditional and marginal relationships; highlight that it is first time we have both tools (here, GJAM), and data (MASTREE+), to test long-standing but untested theories, L67-L69 and L80-L86
- 4th paragraph of the Introduction; we now finish with direct statement of the relevance of our findings, L95-L96
- 4th paragraph of the Discussion; we have expanded it to place the link between masting and stem tissue density in a broader context, L199-L210
- final paragraph of the paper: flesh-out that using the GJAM had dramatic effect on the theory tested. PCA would conclude no effects of conservative tissues on interannual variation in seed production. L229-L231

Besides the way the study is presented, I do have some questions/issues with the method they follow. The first big concern I have is the fact that they only have functional traits for 210 species and had to use machine learning and phylogeny to infer missing data for 307 species. This is problematic to me for a few reasons. 1. Not all traits are phylogenetically

conserved... 2. They have more missing data than they have data.. 3. It seems that at the end the study ends up being circular.

1.2. Thank you for that point. Missing trait data imputation is widely implemented in the functional trait literature (see e.g. Penone et al 2014; Debastiani et al 2021; and with application in large dataset in e.g. Carmona et al 2021a, 2021b, Guillemot et al 2022), perhaps because it's often analyzed with PCA-like analyses that do not allow missing values. In our case, GJAM allows missing observations, so we re-run the model, now without imputation. The results are consistent with the findings reported by the model with data filling (see Table S1 in the supplement, pasted also below).

We revised the text to include the information that GJAM models were run also on data without imputation and that they present similar results, and included the table below in the supplement (Table S1).

Literature

Carmona, C. P., Tamme, R., Pärtel, M., De Bello, F., Brosse, S., Capdevila, P., González, R. M., González-Suárez, M., Salguero-Gómez, R., Vásquez-Valderrama, M., & Toussaint, A. (2021a). Erosion of global functional diversity across the tree of life. *Science Advances*, 7(13), 1–13. <https://doi.org/10.1126/sciadv.abf2675>

Carmona, C. P., Bueno, C. G., Toussaint, A., Träger, S., Díaz, S., Moora, M., Munson, A. D., Pärtel, M., Zobel, M., & Tamme, R. (2021b). Fine-root traits in the global spectrum of plant form and function. *Nature*, 597(7878), 683–687. <https://doi.org/10.1038/s41586-021-03871-y>

Debastiani, V. J., Bastazini, V. A. G., & Pillar, V. D. (2021). Using phylogenetic information to impute missing functional trait values in ecological databases. *Ecological Informatics*, 63(April). <https://doi.org/10.1016/j.ecoinf.2021.101315>

Guillemot, J., Martin-StPaul, N. K., Bulascoschi, L., Poorter, L., Morin, X., Pinho, B. X., le Maire, G., R. L. Bittencourt, P., Oliveira, R. S., Bongers, F., Brouwer, R., Pereira, L., Gonzalez Melo, G. A., Boonman, C. C. F., Brown, K. A., Cerabolini, B. E. L., Niinemets, Ü., Onoda, Y., Schneider, J. V., ... Brancalion, P. H. S. (2022). Small and slow is safe: On the drought tolerance of tropical tree species. *Global Change Biology*, 28(8), 2622–2638. <https://doi.org/10.1111/gcb.16082>

Penone, C., Davidson, A. D., Shoemaker, K. T., Di Marco, M., Rondinini, C., Brooks, T. M., Young, B. E., Graham, C. H., & Costa, G. C. (2014). Imputation of missing data in life-history trait datasets: Which approach performs the best? *Methods in Ecology and Evolution*, 5(9), 961–970. <https://doi.org/10.1111/2041-210X.12232>

Masting metric	Conditional traits	Estimate	SE	2.5%	97.5%	significance
CV						
	LMA	1.06e-03	4.27e-04	2.12e-04	1.89e-03	*
	Seed size	-2.33e-05	8.50e-06	-4.06e-05	-6.90e-06	*
	Leaf N	-1.20e-04	3.97e-03	-7.76e-03	7.63e-03	
	Leaf area	-9.00e-07	1.60e-06	-4.00e-06	2.20e-06	
	Stem density	5.02e-01	1.71e-01	1.78e-01	8.38e-01	*
	Height	4.02e-03	2.15e-03	-2.57e-04	8.22e-03	

Table S1 (for CV) : GJAM-derived conditional relationship between masting metrics (CV and AR1) and functional traits (stem tissue density, seed size, LMA, leaf N, leaf area and plant height) after accounting for the effect of climate and phylogeny. Coefficients are reported with 95%CI, with significant functional trait coefficients in bold. GJAM was used here without functional trait imputation (total count of species with missing traits for LMA = 90 species; seed size = 84 species; Leaf N = 96 species; Leaf area = 111 species; Stem density = 84 species; Height = 51 species).

The second question I have about the method is in the choice of their climatic variables. Using MAT and MAP is easy but is it sufficient? When studying temporal phenomena like masting, why aren't any climate variability measures included in their models? Obviously the costs and benefits of masting will greatly differ in a constant environment vs. a highly variable one.

1.3. Following the comment, we explored whether adding climate variation can improve models fit. Extending the current models with climate variability results in decrease in fit. Models with only climate variability had lower fit than models with annual means (see Table below, and Table S2 in the Supplement). We have now added the information in the paper that such models were explored, and we report the DIC tables in Supplement (L294-L297).

Lack of important climatic variation effects is perhaps not surprising; past studies linked masting variation with climate variation and found no (Koenig and Knops, 2000, Am Nat) or weak (Pearse et al. 2020) relationships. Plants are selected to either amplify weather variability (when EoS are present) or decrease it (when variation provides diseconomies of scale; Kelly 1994 TREE, Kelly et al. 2013 Ecol Lett, Bogdziewicz et al. 2020 Current Biol). In turn, the effects of mean climate are representative of various ways the environment may affect selection for masting. For example, masting may be stronger in boreal and temperate zones due where satiation of consumers is easier, whereas may be less important in tropical zones when seeds may hide from seed consumers by low apparency. We discuss that in the 3rd paragraph of the Discussion.

Climatic predictors in GJAM	DIC
$MAP \times MAT + MAT^2 + MAP^2$	10,997
$MAP \times MAT + MAT^2 + MAP^2 + MAP_{\sigma}$	11,005
$MAP \times MAT_{\sigma} + MAT_{\sigma}^2 + MAP^2 + MAP_{\sigma}$	11,190
$MAP_{\sigma} \times MAT_{\sigma} + MAT_{\sigma}^2 + MAP_{\sigma}^2$	11,314

Table S2 :Joint traits model selection (based on the DIC). GJAM were fitted with different combinations of climate covariates, average species climatic conditions (MAP and MAT) and climate variability (MAP_{σ} and MAT_{σ}). Because of the strong negative correlation between MAT and MAT_{σ} ($cor = -0.82$) we did not include them both in the GJAM models.

Finally, I made several comments throughout the figures which could be greatly improved. I am also confused about the caption and what the figures are showing (for example in Figure 5. caption explains the whiskers are the 95% CI but then some posterior distributions that clearly overlap with 0 are dimmed significantly...). This is pretty confusing and makes me doubt the results overall.

1.4. The figures that present model coefficients and their errors present 95%CI (as per standard). There was a typo in the figure legend that stated 97.5%CI, which is now corrected, apologies. All errors are presented uniformly.

The trait that reviewer points as overlapping zero but dimmed significant is probably seed size. We have not called that significant, but was weakly associated, trying to avoid the strong cut-off at 0.05 that is generally criticized. We now directly add that the negative relationship between seed size and CV is not significant (L133). However, since it is significant in the model without data imputation (see also response 1.2.), we believe that the evidence for rejecting it is weak. Nonetheless, it surely rejects the positive link between CV and seed size, as predicted by the theory, which we discuss in the Discussion (5th paragraph).

We also shortened the paragraph that discusses the link between CV and seed size (see 5th paragraph of Discussion).

In conclusion, I think there is some potential with this study but the authors need to present their work and their results in a more convincing way and rethink (or justify better) some of their analyses/choices.

Introduction

Overall, the introduction doesn't do a very good job setting up the novelty of this and why it is an important study even.

1.5. The introduction was rewritten to better stage the novel contributions coming from our analysis, please see response 1.1.

Results

Since 517 species were analyzed, it would have been interesting to know about variation within the broad life form. For example, comparing results among trees, shrubs, grasses as presented the results are a little too superficial and I would have liked a deeper analyses of the results.

1.6. Thank you for that suggestion. We are unaware of any a priori expectation in the literature according to which masting should vary according to plant life form; or according to which masting-trait / masting-climate associations would be expected to vary by life form. Pearse et al. (2020) included life forms in their analysis across ~300 species, even if without any articulated mechanism, and found that to be not important. However, if the Editor or Reviewer has strong feelings about that matter, we are happy to reconsider.

Nonetheless, we have run additional analysis to make sure we are not missing any major factors. The GJAM results are generalizable across life forms as phylogeny is accounted for in the model. However, we also explored modes where life form was added specifically as a covariate, which has not changed the outcomes.

Finally, we run a functional trait space analyzes to see whether masting syndrome (axis 3 of our PCA related to Axis 1) differs across life forms, and found no support for that (please, see graph below). That figure was added to the Appendix (Figure S2) with distribution of masting metrics for each growth forms.

Fig R1. Trait probability density function for principal components between axis 3 and axis 1 according to plant growth form (other in left panel, shrub in the middle and tree in the right panel). For each growth form groups, the colors indicate the probabilistic distribution of trait combinations in the functional trait space defined by a PCA (ranging from low probability in pale white to high probability in red). Contour lines indicate 0.99, 0.50, and 0.25 quantiles of the probability distribution. We estimated the occurrence probability of a given combination of trait values determined by the principal components axis and bivariate trait combination using two-dimensional kernel density estimation. Analysis and plots have been made with the funspace R package (Carmona et al, 2023).

Litterature

Carmona, C. P., Pavanetto, N., & Puglielli, G. (2023). *funspace : an R package to build , analyze and plot functional trait spaces*. 1–26.
<https://doi.org/https://doi.org/10.1101/2023.03.17.533069>

Line 99. Need to specify what this principal component analyses was (PCA of X functional traits and Y masting metrics).

1.7. Revised as suggested (L101).

Lines 99-114. I am not sure why the authors bother with a simple PCA rather than focusing on the results from GJAM. Need to justify the PCA approach. Also, the authors should spend more time exploring results with GJAM (See my first comment in the results).

1.8. We included the PCA in the paper as it is widely used in functional trait literature. Using PCA allows us to build the contrast between GJAM to an analysis that ignores covariance among traits, climate, and phylogeny. We have revised the text to highlight the difference between a widely used tool (PCA) to the joint trait model (GJAM) (e.g.L80-L85, L98-L100, L108-110 and L230). We have also shortened (by ~50%) the paragraph that describes the results of the PCA (first paragraph of the Results).

As for the first comment in the results, please see the response to that comment 1.6.

Figure 1. The Whittaker diagram is a little strange with many of your points falling outside of the biomes. Use a different diagram? Double check the T and P values of some of these locations? Color code the map with biomes instead in which case you don't need b at all.

1.9. The MAT and MAP in our data are correct. It is not uncommon for the real datasets to fall outside of the model presented by the Whittaker diagram (see e.g. <https://www.nature.com/articles/sdata2018249> or <https://www.nature.com/articles/s41559-021-01471-7>).

The suggestion to categorize the data is interesting. However, If we color-code the points according to the categories, we will reduce our continuous variation into few categories, losing the information on the climatic variation.

Thus, we have kept the figure as is in the revised version, but we could remove the background (the Whittaker biomes) from 1b, if the Editor or the Reviewers have strong feelings about this. We retained it as we feel that it adds information and makes it easier for a reader to put the climatic variation in our data into context. We have revised the legend of Figure 1 to highlight the goal of Fig. 1b is to show the climatic variation across species, whereas the goal of Fig. 1a is to show the spatial location of MASTREE+ location.

Figure 2. Add a clip-art of a species representative of the two extreme masting strategies. Change the point colors (it is too similar to the color picked for the three axes) Instead of plotting Axis.3. vs Axis.2, Plot Axis. 3 vs. Axis.1 so a and b and easily be interpreted together.

1.10. We have replotted the figure as suggested (Axis 3 vs Axis 1), and revised the coloring.

We have not included the clip art of species, as in our case representative forms of extreme masting strategies do not translate into very different life forms that can be summarized in a clip art. While it can work well for PCAs that summarize traits such as size (as in Diaz et al. 2016, Nature), in our case plants of various forms can be found at both ends of Axis 3 (which is highlighted by the fact that masting create a separate axis of trait variation).

Figure 4. Now sure why here you are showing 97.5 CI but then in Figure 5 you can showing 95% CI. I'd be better to remain consistent across results.

1.11. Apologies, it was a typo, corrected. All plots show 95% CI.

Figure 5. Besides a different CI than in Figure 4, I am confused about some of the results. If the whiskers extend to 95%CI, then seed size is not significantly associated with CV or AR1, and leaf N is not significantly associated with AR1.

1.12. Please, see the response 1.4. about the seed size and CV relationship. The relationship between AR1 and leaf N is not mentioned as significant and not discussed in the paper.

Methods

Lines 254-269. You obtained data for 210 species, and used a machine learning algorithm to get data for 517 additional species using phylogeny. This is worrisome to me for several reasons: 1. Many functional traits are not phylogenetically conserved, 2. You end up creating data from more species than you have data for, 3. Then your analyses become circular (phylogeny use to predict traits, then traits and phylogeny use to predict masting after removing climate...). This issue needs to be addressed.

1.13. Please, see response 1.2.

Lines 271. Is it unclear why the authors decided to focus on mean annual T and P only. What about variability? Masting is temporal phenomenon and variability in the climate should be considered.

1.14. Please see response 1.3.

Discussion

Line 213. Again you didn't really tackle the idea of growth form. You have the data but didn't explore it, you stopped at functional traits.

1.15. Please, see response 1.6.

Lines 217-222. How complicated would it be to test this evolutionary model? Could you add it to this paper to make this study stronger?

1.16. We believe that running a well-done evolutionary model is a long-term project and an exciting avenue for another case study.

Referee: 2

The manuscript tests important hypotheses on the evolution of masting in perennial plants. The data and analyses reported are robust, and their results present a mix of confirmations and novel evidence on traditional and so far untested theories. The paper therefore makes an important, although preliminary contribution towards the understanding of the evolution of masting.

2.1. Thank you for your careful evaluation of our paper and suggestions for improvement. Please, see below how we revised the paper.

My only main concern stems from one of the predictors considered, i.e., climate. The question is simple: how can we be sure that current climatologies (extracted by CHELSA) are the same one that acted upon evolution? Also given that climate stability has been higher or lower in different parts of the world (as influenced by eg glaciation, volcanism, plate tectonics...). While the temporal scale for other predictors (phylogeny, life traits) is consistent with evolutionary dynamics, I am afraid of a potential mismatch between these and the climate information used in the paper. I wonder whether using some palaeoclimate data (eg from the PAGES project, or similar) might be useful to answer this question.

2.2. To address this concern, we have re-analyzed our data using the paleo-climate extracted from PALEO-PGEM-Series (Barreto et al. 2023, GEB), taken at three time points (1 million years ago, 2M, and 5M). Using that data does not affect our main conclusions, which is that CV is higher in plants that produce conservative tissues (please, see Figure below).

While we present the results of that additional analysis here, we have not included this result in the main text, for the following reasons. The effect of climate on masting can work in at least two ways. Climate can represent broader environmental contexts (e.g. tropics vs temperate boreal zones) in which masting may be less or more intense due to changes in interactions with seed consumers or pollinators. Climate may also have effects on masting by affecting resource cycles. It is currently unclear what is the primary factor or how do they interact; we discuss this issue in the 3rd paragraph of the Discussion. The Paleo climate is probably a good surrogate for the first, but less so for the latter. At the same time, we are unsure how well the past (e.g. 5M years ago) locations from which climate we derived represent past the distribution of the species that we work on, which probably makes the current climate as a reasonable solution. Nonetheless, if the Reviewer or Editor believes that supplemental analysis should be presented in the text, we are happy to reconsider.

Figure R2. Comparison between MAT/MAP from current (CHELSA) or paleo-climate (1M, 3M and 5M) and conditional relationships among masting metrics and functional traits. a) bi-plot between MAT and MAP. Ellipse indicates climate time (current or paleoclimate, of 5, 3 or 1M years). b,c,d) GJAM-derived, conditional relationship between masting metrics (CV and AR1) and functional traits (stem tissue density, seed mass, LMA, leaf N, leaf area, and plant height) ($n = 517$ species) after accounting for effects of climate and phylogeny. Boxplots show the mean estimate and are bounded by 80%CI with whiskers indicating 95%CI. Colors highlight signs of the correlation (green for positive and purple for negative), with opacity increasing from 80% to 95% of the distribution outside of zero. Grey is for coefficients that overlap zero. GJAM conditional relationship was based on paleo climate in b) climate 5M years, c) climate from 3 M years and d) climate from 1M years.

Literature

Barreto, E., B. Holden, P., R. Edwards, N., & Thiago F., R. (2023). PALEO-PGEM-Series: A spatial time series of the global climate over the last 5 million years (Plio-Pleistocene). *Global Ecology & Biogeography*, 32(7), 1034–1045. <https://doi.org/https://doi.org/10.1111/geb.13683>

Minor comments follow:

1 Conservative tissue construction: not entirely clear from the title what this is about - how about "the need for conservative tissue"?

2.3. Revised as suggested.

15 extreme: not sure this is a proper descriptor for masting... I suggest to stick with "variable"

2.4. Revised as suggested.

16 perennials -> perennial plants

2.5. Revised as suggested.

19 ancestry -> phylogeny? evolutionary history? (also at L213)

2.6. Revised as suggested.

20 was rare -> is rare

2.7. Revised as suggested (see 2.8.).

20-21 it is rare... to lack: replace two negatives with one assertive sentence, e.g., it is common... to include

2.8. Revised as suggested.

23: was -> is

2.9. Revised as suggested.

23: invest in low tissue mortality: not entirely clear... do you refer to species whose tissues are predominantly conserved from year to year (hence the title)? I think this trait is not so familiar to most readers, so a sentence explaining why it was studied and why it is important might benefit clarity

2.10. Great spot, thank you. We revised the 2nd part of the abstract to better flash-out that novel points.

43: remove first names in citations

2.11. Corrected.

50 and rapid utilization: maybe "from" rapid utilization?

2.12. Revised as suggested.

51 such as -> measured by

2.13. Revised as suggested.

63-64: not entirely clear why high stem tissue density would be related to lower mortality rates (eg add ... "due to stronger stress resistance")

2.14. Revised as suggested.

69 and elsewhere: I prefer "stem tissue density" to just "stem density", which might be confused with the number of stems per unit area

2.15. We now uniformly use “stem tissue density” through the text.

73 "if high interannual variation in seed production requires species to produce conservative stems": but the evidence points to the opposite, ie. little masting in dense tropical woods

2.16. Agree, this is just a hypothetical illustration of direct and indirect relationships. Masting CV is low in tropics, where stem tissue density is high. Thus, we can actually find a negative correlation between masting and stem tissue density if we do not account for the climatic covariance. We accounted for climate effects, and found that CV is higher where stem tissue density is high.

We slightly changed the wording in the focal paragraph (L71).

fig 1: why some dots are well out of existing Whittaker biome space?

2.17. Please, see response 1.9.

101: seed mass, tree height and stem (tissue) density can also be expected to vary on separate axes, eg, tall, fast-growing and light-wooded, light-seeded trees. Was this observed on other PCs?

2.18. Yes, it was Axis 5, but we are not reporting all axes since they appear less relevant to the questions posted in this study.

151 "once there is less ecological need for it": losing an evolutionary trait should imply something more than just "no need" for it (in which case it could persist by evolutionary inertia), i.e., it becomes too costly for the new conditions and/or discourages survival and effective reproduction

2.19. Good catch, revised to: “perhaps because species quickly lose their inherited seed production variability once there is no ongoing selection for it”. (L147 and L152, but also in the Abstract, L23).

159-160: this paragraph takes for granted that masting, likely a multi-genetic and very complex trait, is strongly inherited. Authors should make this assumption explicit alongside with appropriate references to support it

2.20. This is a very good point. We have now added in the introduction a reference to Caignard et al. 2019 (L34) that suggest that interannual variation is heritable. We also submitted another paper recently that shows it is the case in *Sorbus aucuparia*; in case the Editor or the Reviewer would feel citing another source is necessary, we can upload that unpublished paper as pre-print and cite it as well.

170-2: oaks are sprouters while pines are not. Maybe there could be a role for disturbance resilience here? see <https://royalsocietypublishing.org/doi/full/10.1098/rstb.2020.0384>

2.20. This is a very exciting idea; we have added a new sentence to the focal paragraph; “One interesting way forward is to examine this question in light of the high resprouting

abilities of oaks but not pines (Vacchiano et al. 2021)". (L169)

173-74: that CV and AR1 are not necessarily linked is new to me. This is an interesting conclusion, maybe authors should elaborate a bit more how and why there can be a high CV without AR1 (no depletion?), or even weirder, a high AR1 with small CV (what would that mean from a mathematical point of view?)

2.21. It is because the highest CV is achieved when there are multiple failure years; such failures after failures increase (make less negative) the values of AR1. Thus, highest CV cannot have extremely negative AR1. Nonetheless, high CV values usually mean consistently negative AR1 (even if not lowest, please see Pearse et al. 2020 *New Phytologist*, Fig. 4 or Dale et al. 2021 *PTRSB*, Fig. 2). In turn, AR1 would be probably strongly negative for perfect alternation (failures and bumper crops always coming one after another).

Nonetheless, for a given value of high CV, there is a considerable variation in AR1 (e.g. 1.5; see our Fig. S5), which we believe is linked to the strength of failure after mast years.

We have added that sentence to the focal paragraph of the discussion (L174): "High CV values without negative AR1 may happen if mast years are not followed by complete failure years (Pearse et al. 2020, Dale et al. 2021)"

207-211: also, the importance of post-germination survival could be invoked here (eg for species with large seeds that are subject to a large survival pressure in the seedling stage)

2.22. Apologies, but we are not sure how that is linked. If the Reviewer would like to elaborate, we are happy to point into that mechanism in the focal section.

212-26: In this "summary", novel information and interesting lessons to be learned are presented, but the text is quite compact. I suggest expanding a bit in a full Conclusion paragraph, so as to have a bit more room to explain implications properly.

2.23. Thank you for that comment. We gave that a few tries, but struggled to find what is the major point that the Reviewer would like to see expanded. We have highlighted the importance of accounting for the effects of climate (L230), but we would be happy to elaborate more if there are specific points that the Reviewer or Editor feels would benefit from expanding.

255+: I am not an expert in this technique. However, would carrying out the analysis on only the 210 species with complete traits have changed the main results?

2.24. Similar concern was shared by Reviewer 1. Analysis of the data without trait imputation (allowed by GJAM) presents qualitatively the same results. Please see response 1.2.

277+: how about testing also extreme temperatures and precipitation rather than their averages, as an indicator of potential environmental stress or disturbances? (as per hypotheses by <https://royalsocietypublishing.org/doi/full/10.1098/rstb.2020.0384>)

2.25. We included that in our models, but the new predictors were not important, please see response 1.3.

278: what is the time window used by CHELSA climatologies? Surely this is not at all comparable to the kind of time span required for selective pressure? In other words, how can we be sure that these are the climates that acted upon evolution? Also given that climate stability has been higher or lower in different parts of the world (as influenced by eg glaciation, volcanism, plate tectonics...)

2.26. Please, see response 2.2.

279 aggregated: you mean by averaging?

2.27. Yes, revised.

280-82: this procedure might be unnecessary, or even accurate, if the observations from mastree would cover a smaller territory than the observations from GBIF, although fewer

2.28. We did this as trait values are species-level means. Of course, it can be argued the other way around, as noticed by the Reviewer. Nonetheless, that choice does not impact the results, which we tested for and added in the focal section (L293): “Nonetheless, MAT and MAP obtained through MASTREE+ sites and GBIF present strong correlations (Fig. S9), and using both provides qualitatively the same results. “

298+: PCA is best for linear relationships, and can be affected by data distortion (horseshoe effect). Have you thought of running a NMDS to account for nonlinear effects, and/or DCA to correct for the arch distortion?

2.29. We have used GJAM to address the shortcomings of the PCA. As the major outcomes and interpretation of relationships comes from the GJAM model but not PCA, we feel that our results are robust against biases in PCA.

We have revised the text to highlight that the PCA is just supporting analysis, with GJAM being the main outcome, please see details at response 1.8.

Reviewers' Comments:

Reviewer #1:

Remarks to the Author:

Thank you for taking the time to revise your manuscript. I really appreciate the work you put into your revisions and the additional analyses you undertook. I have some further questions about the missing traits, and the addition of climatic variability in the model. Please see the attachment for my specific comments.

Referee: 1

General

In this study the authors compile a large dataset of plant functional traits, including masting characteristics, and climate to investigate patterns of masting across vascular plants. The dataset is interesting and understanding why some species exhibit masting and not others is an intriguing and on-going research question. However, it wasn't clear from the set up of this paper what was novel about their study. It wasn't clear why the authors were doing what they were doing. It read more like a data exploration than a novel study (the authors followed the exact methods they previously published with a new dataset, which is not bad but I wish the study was presented to us as a novel exciting study addressing new questions).

1.1. Thank you for that important comment. We believe that the key finding of our work is that variation of seed production correlates with functional traits associated with low mortality. The largest masting cost is skipped reproductive opportunities, at the cost of each missed opportunity should decline as the annual survival increases. That is a fundamental assumption of masting theory that has not been tested at scale. We filled that gap using novel methods that allowed separation of the effects of climate. We believe that the non-standard modeling is key as well, since traditional PCA run on such large scales is confounded by climate heterogeneity.

We have rewritten the Abstract, Introduction, and Discussion of the paper to more directly flesh-out the novel results the study brings. This include revisions of:

Thank you for addressing this point. The novelty is more obvious now. I have no further comment.

Besides the way the study is presented, I do have some questions/issues with the method they follow. The first big concern I have is the fact that they only have functional traits for 210 species and had to use machine learning and phylogeny to infer missing data for 307 species. This is problematic to me for a few reasons. 1. Not all traits are phylogenetically conserved... 2. They have more missing data than they have data.. 3. It seems that at the end the study ends up being circular.

1.2. Thank you for that point. Missing trait data imputation is widely implemented in the functional trait literature (see e.g. Penone et al 2014; Debastiani et al 2021; and with application in large dataset in e.g. Carmona et al 2021a, 2021b, Guillemot et al 2022), perhaps because it's often analyzed with PCA-like analyses that do not allow missing values. In our case, GJAM allows missing observations, so we re-run the model, now without imputation. The results are consistent with the findings reported by the model with data filling (see Table S1 in the supplement, pasted also below).

We revised the text to include the information that GJAM models were run also on data without imputation and that they present similar results, and included the table below in the supplement (Table S1).

Literature

Carmona, C. P., Tamme, R., Pärtel, M., De Bello, F., Brosse, S., Capdevila, P., González, R. M., González-Suárez, M., Salguero-Gómez, R., Vásquez-Valderrama, M., & Toussaint, A. (2021a). Erosion of global functional diversity across the tree of life. *Science Advances*, 7(13), 1–13. <https://doi.org/10.1126/sciadv.abf2675>

Carmona, C. P., Bueno, C. G., Toussaint, A., Träger, S., Díaz, S., Moora, M., Munson, A. D., Pärtel, M., Zobel, M., & Tamme, R. (2021b). Fine-root traits in the global spectrum of plant form and function. *Nature*, 597(7878), 683–687. <https://doi.org/10.1038/s41586-021-03871-y>

Debastiani, V. J., Bastazini, V. A. G., & Pillar, V. D. (2021). Using phylogenetic information to impute missing functional trait values in ecological databases. *Ecological Informatics*, 63(April). <https://doi.org/10.1016/j.ecoinf.2021.101315>

Guillemot, J., Martin-StPaul, N. K., Bulascoschi, L., Poorter, L., Morin, X., Pinho, B. X., le Maire, G., R. L. Bittencourt, P., Oliveira, R. S., Bongers, F., Brouwer, R., Pereira, L., Gonzalez Melo, G. A., Boonman, C. C. F., Brown, K. A., Cerabolini, B. E. L., Niinemets, Ü., Onoda, Y., Schneider, J. V., ... Brancalion, P. H. S. (2022). Small and slow is safe: On the drought tolerance of tropical tree species. *Global Change Biology*, 28(8), 2622–2638. <https://doi.org/10.1111/gcb.16082>

Penone, C., Davidson, A. D., Shoemaker, K. T., Di Marco, M., Rondinini, C., Brooks, T. M., Young, B. E., Graham, C. H., & Costa, G. C. (2014). Imputation of missing data in life-history trait datasets: Which approach performs the best? *Methods in Ecology and Evolution*, 5(9), 961–970. <https://doi.org/10.1111/2041-210X.12232>

Masting metric	Conditional traits	Estimate	SE	2.5%	97.5%	significance
CV						
	LMA	1.06e-03	4.27e-04	2.12e-04	1.89e-03	*
	Seed size	-2.33e-05	8.50e-06	-4.06e-05	-6.90e-06	*
	Leaf N	-1.20e-04	3.97e-03	-7.76e-03	7.63e-03	
	Leaf area	-9.00e-07	1.60e-06	-4.00e-06	2.20e-06	
	Stem density	5.02e-01	1.71e-01	1.78e-01	8.38e-01	*
	Height	4.02e-03	2.15e-03	-2.57e-04	8.22e-03	

Table S1 (for CV) : GJAM-derived conditional relationship between masting metrics (CV and AR1) and functional traits (stem tissue density, seed size, LMA, leaf N, leaf area and plant height) after accounting for the effect of climate and phylogeny. Coefficients are reported with 95%CI, with significant functional trait coefficients in bold. GJAM was used here without functional trait imputation (total count of species with missing traits for LMA = 90 species; seed size = 84 species; Leaf N = 96 species; Leaf area = 111 species; Stem density = 84 species; Height = 51 species).

Thank you for addressing this point, however I still have some questions about it. It looks like the results are consistent for CV (the numbers you show in your response to comments are not the same as the ones from page 32). However, there are not consistent for AR1. This is something that you should recognize and briefly discuss. As I mentioned in my first comment,

not all traits are phylogenetically conserved and the way you infer the missing traits strongly rely on this evolutionary assumption.

The second question I have about the method is in the choice of their climatic variables. Using MAT and MAP is easy but is it sufficient? When studying temporal phenomena like masting, why aren't any climate variability measures included in their models? Obviously the costs and benefits of masting will greatly differ in a constant environment vs. a highly variable one.

1.3. Following the comment, we explored whether adding climate variation can improve models fit. Extending the current models with climate variability results in decrease in fit. Models with only climate variability had lower fit than models with annual means (see Table below, and Table S2 in the Supplement). We have now added the information in the paper that such models were explored, and we report the DIC tables in Supplement (L294-L297).

Lack of important climatic variation effects is perhaps not surprising; past studies linked masting variation with climate variation and found no (Koenig and Knops, 2000, Am Nat) or weak (Pearse et al. 2020) relationships. Plants are selected to either amplify weather variability (when EoS are present) or decrease it (when variation provides diseconomies of scale; Kelly 1994 TREE, Kelly et al. 2013 Ecol Lett, Bogdziewicz et al. 2020 Current Biol). In turn, the effects of mean climate are representative of various ways the environment may affect selection for masting. For example, masting may be stronger in boreal and temperate zones due where satiation of consumers is easier, whereas may be less important in tropical zones when seeds may hide from seed consumers by low apparency. We discuss that in the 3rd paragraph of the Discussion.

Climatic predictors in GJAM	DIC
$MAP \times MAT + MAT^2 + MAP^2$	10,997
$MAP \times MAT + MAT^2 + MAP^2 + MAP_{\sigma}$	11,005
$MAP \times MAT_{\sigma} + MAT_{\sigma}^2 + MAP^2 + MAP_{\sigma}$	11,190
$MAP_{\sigma} \times MAT_{\sigma} + MAT_{\sigma}^2 + MAP_{\sigma}^2$	11,314

Table S2 :Joint traits model selection (based on the DIC). GJAM were fitted with different combinations of climate covariates, average species climatic conditions (MAP and MAT) and climate variability (MAP_{σ} and MAT_{σ}). Because of the strong negative correlation between MAT and MAT_{σ} ($cor = -0.82$) we did not include them both in the GJAM models.

Thank you for addressing this point and testing new models. I don't fully follow the logic for why you tested some models and not others. For example, biologically, why is a model with the interaction between MAP and MAT_{σ} more relevant than a model with MAP and MAP_{σ} model? Having high precipitation all the time is different from having high mean precipitation with strong seasonality. Rather than testing model with the quadratic form of variance, you should test more model where variance and mean interact (at least for precipitation). For temperature, I understand there was high negative correlation, but you could test for a model that has $MAT \times MAT_{\sigma}$.

Finally, I made several comments throughout the figures which could be greatly improved. I am also confused about the caption and what the figures are showing (for example in Figure 5. caption explains the whiskers are the 95% CI but then some posterior distributions that clearly overlap with 0 are dimmed significantly...). This is pretty confusing and makes me doubt the results overall.

1.4. The figures that present model coefficients and their errors present 95%CI (as per standard). There was a typo in the figure legend that stated 97.5%CI, which is now corrected, apologies. All errors are presented uniformly.

The trait that reviewer points as overlapping zero but dimmed significant is probably seed size. We have not called that significant, but was weakly associated, trying to avoid the strong cut-off at 0.05 that is generally criticized. We now directly add that the negative relationship between seed size and CV is not significant (L133). However, since it is significant in the model without data imputation (see also response 1.2.), we believe that the evidence for rejecting it is weak. Nonetheless, it surely rejects the positive link between CV and seed size, as predicted by the theory, which we discuss in the Discussion (5th paragraph).

We also shortened the paragraph that discusses the link between CV and seed size (see 5th paragraph of Discussion).

Thank you for clarifying. It makes more sense now and I have no further comment.

In conclusion, I think there is some potential with this study but the authors need to present their work and their results in a more convincing way and rethink (or justify better) some of their analyses/choices.

Introduction

Overall, the introduction doesn't do a very good job setting up the novelty of this and why it is an important study even.

1.5. The introduction was rewritten to better stage the novel contributions coming from our analysis, please see response 1.1.

Thank you

Results

Since 517 species were analyzed, it would have been interesting to know about variation within the broad life form. For example, comparing results among trees, shrubs, grasses as presented the results are a little too superficial and I would have liked a deeper analyses of the results.

1.6. Thank you for that suggestion. We are unaware of any a priori expectation in the literature according to which masting should vary according to plant life form; or according to which masting-trait / masting-climate associations would be expected to vary by life form. Pearse et al. (2020) included life forms in their analysis across ~300 species, even if without any articulated mechanism, and found that to be not important. However, if the Editor or Reviewer has strong feelings about that matter, we are happy to reconsider.

Nonetheless, we have run additional analysis to make sure we are not missing any major factors. The GJAM results are generalizable across life forms as phylogeny is accounted for in the model. However, we also explored modes where life form was added specifically as a covariate, which has not changed the outcomes.

Finally, we run a functional trait space analyzes to see whether masting syndrome (axis 3 of our PCA related to Axis 1) differs across life forms, and found no support for that (please, see graph below). That figure was added to the Appendix (Figure S2) with distribution of masting metrics for each growth forms.

Fig R1. Trait probability density function for principal components between axis 3 and axis 1 according to plant growth form (other in left panel, shrub in the middle and tree in the right panel). For each growth form groups, the colors indicate the probabilistic distribution of trait combinations in the functional trait space defined by a PCA (ranging from low probability in pale white to high probability in red). Contour lines indicate 0.99, 0.50, and 0.25 quantiles of the probability distribution. We estimated the occurrence probability of a given combination of trait values determined by the principal components axis and bivariate trait combination using two-dimensional kernel density estimation. Analysis and plots have been made with the funspace R package (Carmona et al, 2023).

Litterature

Carmona, C. P., Pavanetto, N., & Puglielli, G. (2023). *funspace : an R package to build , analyze and plot functional trait spaces*. 1–26. <https://doi.org/https://doi.org/10.1101/2023.03.17.533069>

Great! Thank you for looking into it and adding this analysis in.

Line 99. Need to specify what this principal component analyses was (PCA of X functional traits and Y masting metrics).

1.7. Revised as suggested (L101).

Thank you

Lines 99-114. I am not sure why the authors bother with a simple PCA rather than focusing on the results from GJAM. Need to justify the PCA approach. Also, the authors should spend more time exploring results with GJAM (See my first comment in the results).

1.8. We included the PCA in the paper as it is widely used in functional trait literature. Using PCA allows us to build the contrast between GJAM to an analysis that ignores covariance among traits, climate, and phylogeny. We have revised the text to highlight the difference between a widely used tool (PCA) to the joint trait model (GJAM) (e.g.L80-L85, L98-L100, L108-110 and L230). We have also shortened (by ~50%) the paragraph that describes the results of the PCA (first paragraph of the Results).

As for the first comment in the results, please see the response to that comment 1.6.

Thank you

Figure 1. The Whittaker diagram is a little strange with many of your points falling outside of the biomes. Use a different diagram? Double check the T and P values of some of these locations? Color code the map with biomes instead in which case you don't need b at all.

1.9. The MAT and MAP in our data are correct. It is not uncommon for the real datasets to fall outside of the model presented by the Whittaker diagram (see e.g. <https://www.nature.com/articles/sdata2018249> or <https://www.nature.com/articles/s41559-021-01471-7>).

The suggestion to categorize the data is interesting. However, If we color-code the points according to the categories, we will reduce our continuous variation into few categories, losing the information on the climatic variation.

Thus, we have kept the figure as is in the revised version, but we could remove the background (the Whittaker biomes) from 1b, if the Editor or the Reviewers have strong feelings about this. We retained it as we feel that it adds information and makes it easier for a reader to put the climatic variation in our data into context. We have revised the legend of Figure 1 to highlight the goal of Fig. 1b is to show the climatic variation across species, whereas the goal of Fig. 1a is to show the spatial location of MASTREE+ location.

I see what you mean, but I do think that for a first figure, it is okay to limit it to the map (showing the breath of the data) and adding the "climate space" in color dots. You have the information and the range of MAT and MAP in your text, and you could keep the whittaker diagram in your appendix. Having a two panel figures for Figure 1, just makes it harder to see the map (which to me is the most valuable).

Figure 2. Add a clip-art of a species representative of the two extreme masting strategies. Change the point colors (it is too similar to the color picked for the three axes) Instead of plotting Axis.3. vs Axis.2, Plot Axis. 3 vs. Axis.1 so a and b and easily be interpreted together.

1.10. We have replotted the figure as suggested (Axis 3 vs Axis 1), and revised the coloring.

We have not included the clip art of species, as in our case representative forms of extreme masting strategies do not translate into very different life forms that can be summarized in a clip art. While it can work well for PCAs that summarize traits such as size (as in Diaz et al. 2016, Nature), in our case plants of various forms can be found at both ends of Axis 3 (which is highlighted by the fact that masting create a separate axis of trait variation).

I understand! Thank you!

Figure 4. Now sure why here you are showing 97.5 CI but then in Figure 5 you can showing 95% CI. I'd be better to remain consistent across results.

1.11. Apologies, it was a typo, corrected. All plots show 95% CI.

Thank you!

Figure 5. Besides a different CI than in Figure 4, I am confused about some of the results. If the whiskers extend to 95%CI, then seed size is not significantly associated with CV or AR1, and leaf N is not significantly associated with AR1.

1.12. Please, see the response 1.4. about the seed size and CV relationship. The relationship between AR1 and leaf N is not mentioned as significant and not discussed in the paper.

Thank you!

Methods

Lines 254-269. You obtained data for 210 species, and used a machine learning algorithm to get data for 517 additional species using phylogeny. This is worrisome to me for several reasons: 1. Many functional traits are not phylogenetically conserved, 2. You end up creating data from more species than you have data for, 3. Then your analyses become circular (phylogeny use to predict traits, then traits and phylogeny use to predict masting after removing climate...). This issue needs to be addressed.

1.13. Please, see response 1.2.

Thank you but see my further questions.

Lines 271. Is it unclear why the authors decided to focus on mean annual T and P only. What about variability? Masting is temporal phenomenon and variability in the climate should be considered.

1.14. Please see response 1.3.

Thank you but see my further questions.

Discussion

Line 213. Again you didn't really tackle the idea of growth form. You have the data but didn't explore it, you stopped at functional traits.

1.15. Please, see response 1.6.

Thank you!

Lines 217-222. How complicated would it be to test this evolutionary model? Could you add it to this paper to make this study stronger?

1.16. We believe that running a well-done evolutionary model is a long-term project and an exciting avenue for another case study.

Thank you!

Reviewer #2:

Remarks to the Author:

I thank the authors for a very thorough and complete review of all concerns raised. The paper now looks a robust and meaningful contribution to the existing literature

REVIEWER COMMENTS TO AUTHOR

Referee: 1

Thank you for addressing this point, however I still have some questions about it. It looks like the results are consistent for CV (the numbers you show in your response to comments are not the same as the ones from page 32). However, there are not consistent for AR1. This is something that you should recognize and briefly discuss. As I mentioned in my first comment, not all traits are phylogenetically conserved and the way you infer the missing traits strongly rely on this evolutionary assumption.

1.1. Thank you for pointing this out. We now mention the AR1 relationships with leaf N and height in the text (L259-264).

Thank you for addressing this point and testing new models. I don't fully follow the logic for why you tested some models and not others. For example, biologically, why is a model with the interaction between MAP and MAT_var more relevant than a model with MAP and MAP_var model? Having high precipitation all the time is different from having high mean precipitation with strong seasonality. Rather than testing model with the quadratic form of variance, you should test more model where variance and mean interact (at least for precipitation). For temperature, I understand there was high negative correlation, but you could test for a model that has MAT x MAT_var.

1.2. Thank you for that comment. We have strongly extended the list of models tested in the Table S2, including models specifically asked for by the Reviewer: i.e. MAP_mean x MAP_var, as well as MAT_mean x MAT_var.

I see what you mean, but I do think that for a first figure, it is okay to limit it to the map (showing the breath of the data) and adding the "climate space" in color dots. You have the information and the range of MAT and MAP in your text, and you could keep the whittaker diagram in your appendix. Having a two panel figures for Figure 1, just makes it harder to see the map (which to me is the most valuable).

1.3. To make sure the map is better visible, we have resized the figures on that panel. The map and Whittaker biome were previously 1:1, which is now changed to $\frac{2}{3}$ for map and $\frac{1}{3}$ for the Whittaker. We have also decreased the size of points, increasing the visibility of locations. This indeed looks much better, thank you!

Referee: 2

I thank the authors for a very thorough and compete review of all concerns raised. The paper now looks a robust and meaningful contribution to the masting literature.

2.1. Thank you!

Reviewers' Comments:

Reviewer #1:

Remarks to the Author:

Thank you for addressing my last comments! The first figure looks much better and I appreciate the time you took to explore more deeply climate variability. I understand the fit was not improved with the addition of climate variability, but the fit is only one aspect of developing a model. Besides a lower fit, what relationship did these models identified? Were any climate variability measures significantly associated with masting metrics? Or were climate variability measures never significantly associated with masting metrics?

REVIEWER COMMENTS TO AUTHOR

Referee: 1

Thank you for addressing my last comments! The first figure looks much better and I appreciate the time you took to explore more deeply climate variability. I understand the fit was not improved with the addition of climate variability, but the fit is only one aspect of developing a model. Besides a lower fit, what relationship did these models identified? Were any climate variability measures significantly associated with masting metrics? Or were climate variability measures never significantly associated with masting metrics?

Our response: The model that included variability in mean annual precipitation (MAP) was the model with the second best fit, according to the DIC (see Supp Table 2, copied below). In that model, the 95% credible intervals of the effect of MAP variability on both AR1 and CV overlapped with 0.

The sixth-best model included the MAT variability. In that case, the 95% credible intervals of MAT variability on CV and AR1 also included 0.

The fit of the second-best model (with MAP variability) scored 8 points lower according to DIC compared to the best model. Such a model, according to the standard model selection criteria, has weak support (Burnham and Anderson 2002). The other models that included either MAT or MAP variability had the Δ DIC less than 10, which means that these model fits received essentially no support. In other words, the probability that one of the alternative models is the best for the data is 0 (Burnham and Anderson 2002).

On this basis, unless advised otherwise by the editor, we would prefer not to revise the manuscript to include any further discussion of these results, noting that the model outputs are included in the supplementary information already, and the analysis is described at L278-291 in the main text.

Table S2: Joint traits model selection (based on the DIC values). GJAM models were fitted with different combinations of climate covariates, average species climatic conditions (MAP and MAT) and climate variability (MAP_{σ} and MAT_{σ}). Some model combinations were excluded due to collinearity issues.

Climatic predictors in GJAM	DIC
$MAP \times MAT + MAT^2 + MAP^2$	10,997
$MAP \times MAT + MAT^2 + MAP^2 + MAP_{\sigma}$	11,005
$MAP \times MAT$	11,063
$MAP_{\sigma} \times MAT + MAT^2 + MAP_{\sigma}^2$	11,113
$MAP_{\sigma} \times MAT$	11,149
$MAT_{\sigma} \times MAT$	11,159
$MAP \times MAT_{\sigma} + MAT_{\sigma}^2 + MAP^2 + MAP_{\sigma}$	11,190
$MAP_{\sigma} \times MAT_{\sigma} + MAT_{\sigma}^2 + MAP_{\sigma}^2$	11,314
$MAP_{\sigma} \times MAP + MAP^2 + MAP_{\sigma}^2$	11,464
$MAP \times MAT_{\sigma}$	11,505
$MAP \times MAP_{\sigma}$	11,576
$MAP_{\sigma} \times MAT_{\sigma}$	11,652